Ecological and Evolutionary Science
# Strain-Level Diversity Impacts Cheese Rind Microbiome Assembly and Function

Brittany A. Niccum,[a] Erik K. Kastman,[a] Nicole Kfoury,[b] Albert Robbat, Jr.,[b] ◉ Benjamin E. Wolfe[a]

[a]Tufts University, Department of Biology, Medford, Massachusetts, USA
[b]Tufts University, Department of Chemistry, Medford, Massachusetts, USA

**ABSTRACT** Diversification can generate genomic and phenotypic strain-level diversity within microbial species. This microdiversity is widely recognized in populations, but the community-level consequences of microbial strain-level diversity are poorly characterized. Using the cheese rind model system, we tested whether strain diversity across microbiomes from distinct geographic regions impacts assembly dynamics and functional outputs. We first isolated the same three bacterial species (*Staphylococcus equorum*, *Brevibacterium auranticum*, and *Brachybacterium alimentarium*) from nine cheeses produced in different regions of the United States and Europe to construct nine synthetic microbial communities consisting of distinct strains of the same three bacterial species. Comparative genomics identified distinct phylogenetic clusters and significant variation in genome content across the nine synthetic communities. When we assembled each synthetic community with initially identical compositions, community structure diverged over time, resulting in communities with different dominant taxa. The taxonomically identical communities showed differing responses to abiotic (high salt) and biotic (the fungus *Penicillium*) perturbations, with some communities showing no response and others substantially shifting in composition. Functional differences were also observed across the nine communities, with significant variation in pigment production (light yellow to orange) and in composition of volatile organic compound profiles emitted from the rinds (nutty to sulfury).

**IMPORTANCE** Our work demonstrated that the specific microbial strains used to construct a microbiome could impact the species composition, perturbation responses, and functional outputs of that system. These findings suggest that 16S rRNA gene taxonomic profiles alone may have limited potential to predict the dynamics of microbial communities because they usually do not capture strain-level diversity. Observations from our synthetic communities also suggest that strain-level diversity has the potential to drive variability in the aesthetics and quality of surface-ripened cheeses.

**KEYWORDS** cheese, genomics, microbial communities, microbiome assembly, strain diversity

An ongoing challenge in the study of microbial community diversity is to move from describing patterns of diversity to identifying processes that generate diversity. Microbial ecologists have adopted a conceptual framework that considers the relative roles of dispersal, selection, drift, and diversification in microbiome assembly (1–3). Through both observational studies of *in situ* microbial communities and experimental manipulations of field and laboratory microbiomes, many studies have focused on the short-term ecological processes of dispersal and abiotic and biotic selection (1, 4). The impacts of the longer-term evolutionary process of diversification on microbiome assembly have rarely been experimentally assessed (5).

In the present work, diversification is defined as the emergence of genetic variation

Address correspondence to Benjamin E. Wolfe, benjamin.wolfe@tufts.edu.

Strain-level diversity impacts how cheese rind microbiomes develop. This work may explain why cheeses with similar microbes can have vastly different aromas.

within microbial populations (1). This phenomenon of within-species genetic diversity and associated phenotypic diversity is often called strain-level diversity or microdiversity in microbiology (6–9). Experimental and observational studies have demonstrated that microbial populations can rapidly diversify as they adapt to local selection pressures through mutation and recombination. Studies using experimental evolution of laboratory populations have demonstrated how diverse strains that vary in ecologically important traits can rapidly evolve from genetically and phenotypically homogenous populations (10–14). In natural populations, population-level comparative genomic studies and strain-resolved metagenomic surveys have revealed extensive genomic and phenotypic variation within microbial species (15–20). These studies have clearly demonstrated the origins and extent of genomic and phenotypic diversity within microbial taxa. The ecological significance of this strain-level diversification in a multispecies community context is still largely unknown.

One prediction is that strain-level diversity observed in microbial monocultures could have little impact on multispecies microbiome assembly and functioning if pairwise interactions or functional redundancy within the community buffers genomic and phenotypic variation. For example, the community-level impact of strain-level variation in a specific nutrient uptake pathway observed in one species could be swamped out by strong provisioning or resource competition of the same nutrient by a neighboring species. Alternatively, strain-level variation may have strong impacts on community assembly and function if variation in ecological traits is strong enough to alter pairwise interactions or dynamics of succession within a community. For example, strain-level diversity in nutrient uptake of an early colonizing species could impact successional dynamics of the entire community if it substantially impacts nutrients available for other community members. We are unaware of studies that experimentally address either of these predictions in microbial communities. Studies in plant communities have demonstrated that intraspecific genetic and phenotypic diversity can impact community assembly and function (21). Whether diversification of microbial taxa has similar community-level consequences has not been experimentally tested.

Cheese rinds provide an ideal opportunity to test the ecological significance of strain-level diversity within microbiomes. Rinds form on the surfaces of cheeses aged in an aerobic environment and are composed of bacteria, yeasts, and filamentous fungi (22–24). Our previous work used amplicon and shotgun metagenomics to describe the bacterial and fungal diversity of 137 cheese rinds from the United States and Europe (22). Three bacterial genera—*Staphylococcus*, *Brevibacterium*, and *Brachybacterium*— were the most frequently detected across cheese rinds (*Staphylococcus* was detected in 78.5% of cheeses, *Brevibacterium* in 66.9%, and *Brachybacterium* in 68.8%). Through variations in abiotic (salinity, pH, resource availability) and biotic (presence of bacterial and fungal neighbors) selection pressures applied during cheese production and aging (25), cheese microbiomes have the potential to evolve new genotypes and phenotypes with divergent functions. Some studies have characterized the strain diversity of individual cheese rind microbes for technological applications (26–36), but whether this strain diversity impacts ecological interactions and community development of cheese rind microbial communities has not been examined.

Here, we characterized strain-level diversity across cheese rinds and determined its consequences for community assembly and function. We isolated the same three species of bacteria—*Staphylococcus equorum* (hereafter *Staphylococcus*), *Brevibacterium auranticum* (hereafter *Brevibacterium*), and *Brachybacterium alimentarium* (hereafter *Brachybacterium*)—from nine different cheeses made across the United States and Europe (Fig. 1). The three taxa represent the most common species of the three most abundant bacterial genera in cheese rinds (22, 25). *Staphylococcus*, *Brevibacterium*, and *Brachybacterium* can enter the dairy environment from the raw milk used for cheese production and therefore have the potential to co-occur and adapt to abiotic and biotic conditions within local cheese production facilities (37–39). Each species has a distinct colony morphology (Fig. 1B), making it easy to track composition in experimental communities. By inoculating these bacterial strains onto a cheese curd agar (CCA)

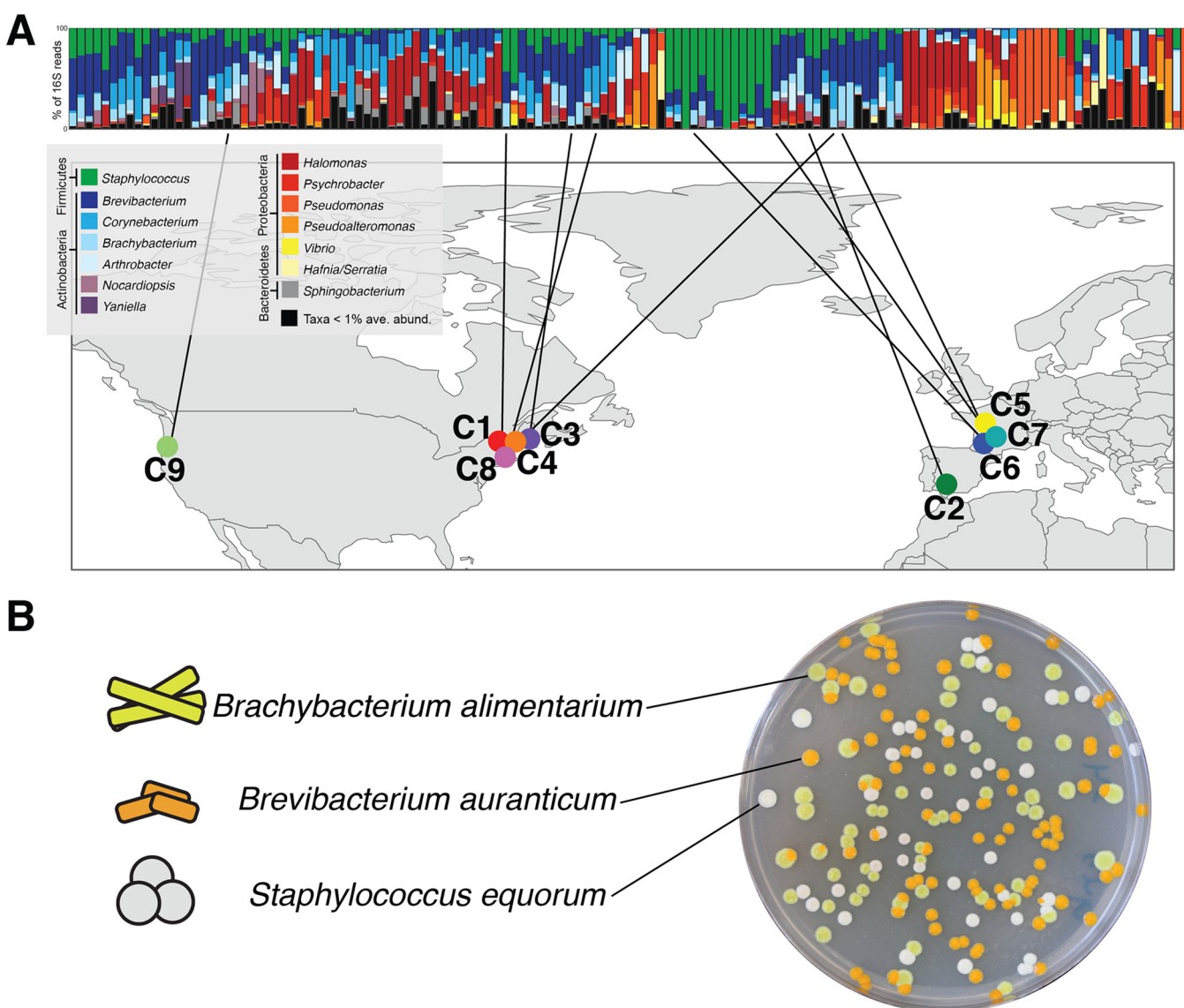

**FIG 1** Isolation of nine synthetic cheese rind communities with distinct strains of the same species. (A) The same three bacterial species—*Brachybacterium alimentarium*, *Brevibacterium auranticum*, and *Staphylococcus equorum*—were isolated from a set of 137 cheese rinds that were previously described using 16S rRNA gene amplicon sequencing (22). Each column represents average relative abundance data for one cheese rind microbiome. (B) The three microbiome species have distinct colony morphologies.

medium, we constructed synthetic cheese rind microbial communities that mimic the microbial dynamics and functions of real surface-ripened cheese rinds. We predicted that intraspecific variation of microbiome members across cheese rind communities would cause differences in synthetic cheese rind microbial community structure. We also predicted that strain-level diversity would result in differences in community functions, including pigmentation of the cheese rind biofilm and the production of aroma compounds.

## RESULTS

**Genomic diversity across nine synthetic cheese rind communities with distinct strains of the same three species.** To determine genomic variation across the nine synthetic cheese rind communities, we constructed draft genomes of each strain (Table S1 in the supplemental material). We used single-nucleotide polymorphisms (SNPs) in the core genes shared across all nine communities to determine phylogenomic divergence of each of the communities (40). We then determined variation in

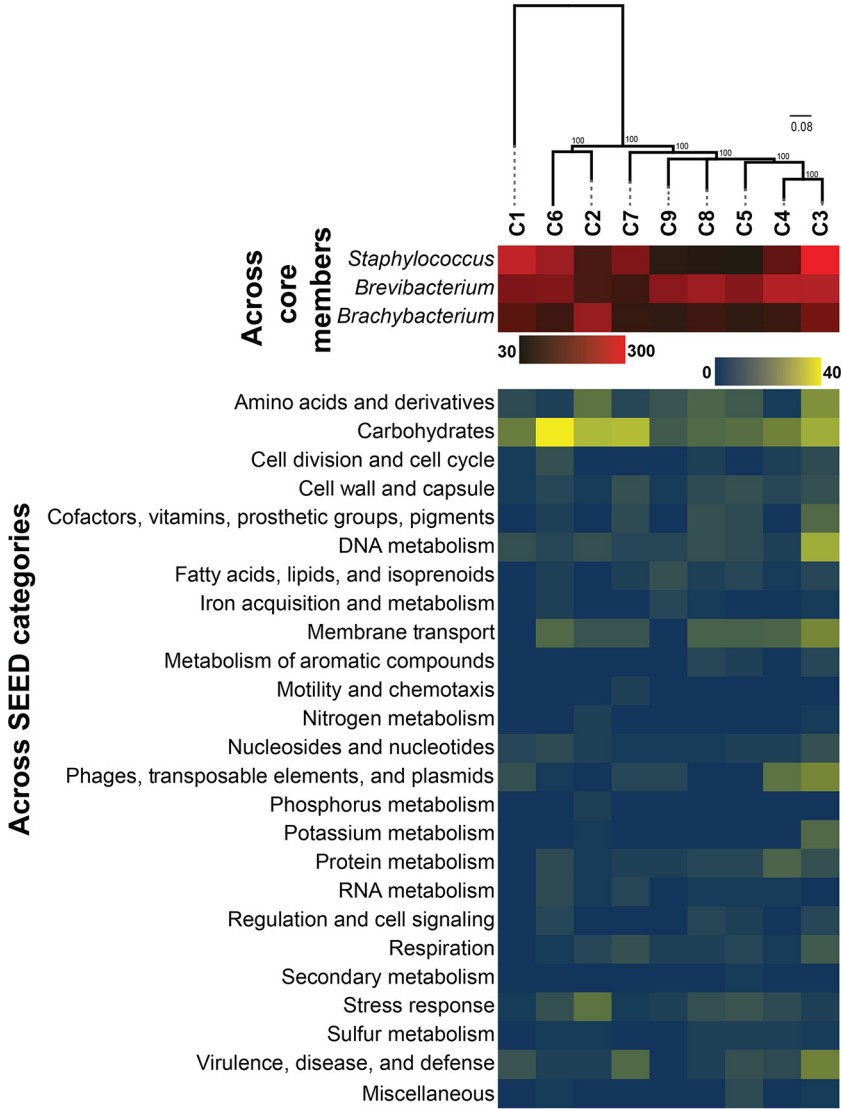

**FIG 2** Accessory genome of the nine synthetic cheese rind communities. The heat map indicates variations in the abundances of unique accessory gene clusters across the three individual taxa (top) and across SEED functional categories (bottom). Phylogeny is shown with a maximum likelihood consensus tree constructed from SNPs identified across the nine communities. Values represent bootstrap support.

functional gene content across the nine communities using Pan-Genome Analysis Pipeline (PGAP) (41). For functional gene content analysis, we focused on accessory genes that were uniquely present in only one community as these genomic traits may help drive divergence in microbiome functions.

The 16S rRNA gene region had 100% pairwise identity within all *Brachybacterium* and *Staphylococcus* strains and 99.9% pairwise identity across the *Brevibacterium* strains (see Fig. S1 in the supplemental material). Across the nine communities, 8,069 gene clusters were shared among all three species, making up the core metagenome of these communities. Using SNPs identified in this core metagenome with PanSeq, phylogenomic divergence across the nine cheese communities was apparent (Fig. 2). C1 was distant from the other eight rind microbiomes, driven by the highly divergent *Staphylococcus* genome in this community (Fig. S2). The other eight rind microbiomes clustered into two broad phylogenomic groups: one containing C6 and C2 and the other containing the remaining six communities (Fig. 2). The total number of unique accessory gene clusters across the nine communities was highly variable, ranging from

mSystems®

246 genes (C5) to 630 genes (C3) (Fig. 2; see also Table S2). Variability in the abundance of accessory gene clusters was most prominent in *Staphylococcus* (ranging from 36 to 280 unique gene clusters across strains) and *Brevibacterium* (ranging from 72 to 213 unique gene clusters), suggesting that these taxa have the most dynamic accessory gene content.

Several biological processes were significantly enriched in the rind communities (Table S3). C3 had the most diverse enrichment of SEED categories, with overrepresentation of genes in potassium metabolism, carbohydrates, and DNA metabolism. Protein metabolism and phages/prophages/transposable elements/plasmids were overrepresented in C4. In C2, the accessory genome was significantly enriched with stress response genes. Carbohydrate-related genes were enriched in the C6 rind microbiome. Some of these unique accessory genes might be functionally significant in the cheese rind environment. For example, *Brevibacterium* of C3 has a unique potassium transport system encoded by genes with high similarity to the *kdfABCF* operon (Table S2) that is known to play a role in salt stress in bacteria (42).

Collectively, these genomic data demonstrate that our nine synthetic cheese communities isolated from distinct cheeses are phylogenomically diverse and have variable genome content. Although the presence/absence of genes does not indicate actual functional potential of microbes, these comparative genomic data suggest that there could be divergence in how each taxon functions within each community and how they respond to perturbations.

**Community assembly dynamics vary across synthetic cheese communities with distinct strains of the same three species.** We next determined whether strain-level differences impacted how the cheese rind communities assemble. A typical community succession in our lab model involves the following steps: (i) early colonization of *Staphylococcus* that can tolerate the low pH (5.0 to 5.2) of the cheese curd, (ii) growth of *Brachybacterium* in the middle step of succession, and (iii) dominance by *Brevibacterium* at the end of succession (22, 43). We predicted two different potential impacts of strain-level variation on community assembly. In one scenario, distinct strains of *Staphylococcus*, *Brachybacterium*, and *Brevibacterium* across the nine communities might vary in genome content or growth rates in isolation, but these differences might be too minor to impact the dynamics of assembly of the three-member community. In this case, we expected nearly identical forms of community composition across the different cheese rind microbiomes as the strains of each species behaved similarly. Alternatively, strain-level differences might translate into differences in interactions with other community members or rates of growth within the community succession. In this scenario, we expected to observe reproducible changes in the composition of the communities as they assembled and differences in functional outputs.

To determine how strain-level differences across communities impact assembly dynamics, we used *in vitro* community assembly assays to measure total CFU and community composition (relative abundance of each species) (Fig. 3A). Communities were quantified at 3 and 10 days after inoculation of equal amounts of each of the three bacterial species on the surface of cheese curd agar. Our previous work demonstrated that this assay mimics *in situ* community dynamics (22, 43). We acknowledge that real cheese rind communities would develop over much longer time scales (weeks to months). In the context of this work, we used the community assembly assay in a standardized environment to demonstrate the potential for divergence in community assembly.

At both 3 and 10 days of community assembly, there were nearly no differences in total community abundance as measured by combined CFU counts of all three species (Fig. 3B) (day 3 analysis of variance [ANOVA] $F_{8,81} = 2.07$, $P = 0.05$; day 10 ANOVA $F_{8,79} = 0.46$, $P = 0.88$). However, there were substantial differences in community composition across the nine cheese rind communities (day 3 permutational multivariate analysis of variance [PERMANOVA] $F = 4.005$, $P < 0.001$; day 10 PERMANOVA $F = 5.57$, $P < 0.001$). Many communities (C1, C2, C6, and C7) were dominated by *Brevibacterium* at the end of succession (Fig. 3C). Some communities (C3, C5, C8, and C9) had a

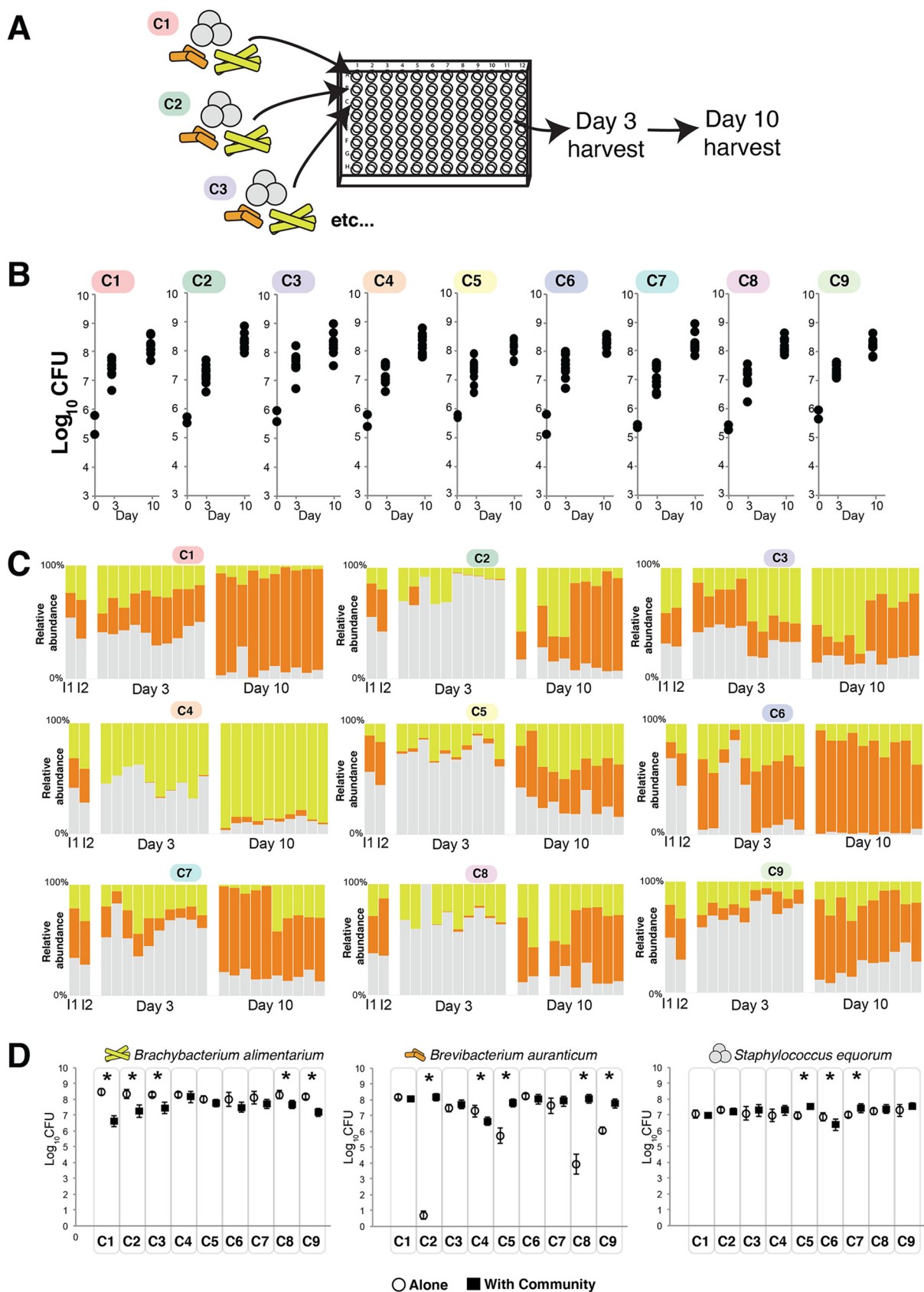

**FIG 3** Divergent community assembly across nine synthetic cheese rind communities with identical species compositions. (A) Experimental setup. Each set of three species from each microbiome was inoculated into wells of 96-well plates. Communities were harvested 3 and 10 days

relatively even mix of all three species. Community C4 had a highly dissimilar structure with a high abundance of *Brachybacterium* at the end of succession and a low abundance of *Brevibacterium*.

A simple explanation for differences in community composition across the nine cheese rind communities is that individual bacterial strains have different growth abilities alone and in the community. Those taxa and strains that grow best alone and with the community present should be the most abundant members of the community. To test this, we determined total growth of each of the 27 strains on cheese curd agar and compared growth alone after 10 days to growth in the community. All *Staphylococcus* species grew well alone and had limited responses to growth in the community (Fig. 3D). Two strains were slightly stimulated by growth in the community (C5 and C7), and one was slightly inhibited (C6). In contrast to the relatively even growth levels of the *Staphylococcus*, the *Brevibacterium* strains had variable growth levels alone across the nine cheese rind communities. Four of the *Brevibacterium* strains (C2, C5, C8, and C9) grew poorly alone on cheese curd agar and were strongly stimulated by growth in the community. One *Brevibacterium* strain (C4) was inhibited by growth in the community. All *Brachybacterium* strains grew well on cheese curd by themselves and were generally inhibited when grown in the community.

For all three taxa, mean growth alone was a very poor predictor of mean relative abundance in the community (*Staphylococcus* $r^2 = 0.166$, $P = 0.276$; *Brevibacterium* $r^2 = 0.001$, $P = 0.923$; *Brachybacterium* $r^2 = 0.020$, $P = 0.716$). A somewhat better predictor of mean relative abundance was how growth of each strain was impacted by the community (*Staphylococcus* $r^2 = 0.672$, $P < 0.01$; *Brevibacterium* $r^2 = 0.013$, $P = 0.773$; *Brachybacterium* $r^2 = 0.319$, $P = 0.113$). This suggests that interactions between each of the strains and their communities may contribute to differences in community composition across the nine synthetic communities. For example, the inhibition of *Brevibacterium* and lack of inhibition of *Brachybacterium* in C4 may partly explain why this community was the only one to be dominated by *Brachybacterium*.

**Strain-level diversity affects cheese rind community responses to abiotic and biotic perturbations.** Cheese rind microbiomes may experience abiotic or biotic perturbations that could alter community assembly and function. We predicted that if strains have evolved different responses to stress or if the communities have coevolved stress-response mechanisms, taxonomically identical experimental communities might have divergent responses to perturbations. Two major perturbations in cheese rind microbiomes are those involving salt and interactions with fungi (22, 25, 44). Salt concentrations are initially high on the surface of fresh cheese because salt is applied to the cheese surface alone or in a brine (45). The salt diffuses into the cheese and eventually equilibrates to a level of around 3% in the rind environment of many cheeses. Cheese rind microbiomes also experience interactions with fungi, ranging from yeasts (e.g., *Debaryomyces* and *Galactomyces* species) to molds (*Fusarium*, *Scopulariopsis*, and *Penicillium* species) (22, 25, 46). *Penicillium* species are widespread in cheese rinds and can strongly inhibit diverse cheese rind bacteria (22, 43, 47), potentially through the production of secondary metabolites or other mechanisms.

To determine how the nine synthetic cheese rind communities would respond to salt and fungal perturbations, we used the community assembly assay described above with the addition of two treatments: a 6% NaCl treatment and a treatment with added *Penicillium* (+*Penicillium*). We used a strain of *Penicillium* that was isolated from a

**FIG 3** Legend (Continued)
after inoculation. (B) Total community abundance as measured by CFU of each of the nine microbiomes. $n = 5$ across two experimental replicates. For the day 0 inputs, only two points are shown to represent the inputs for each of the two experimental replicates. (C) Relative abundance of each of the three bacterial species across each of the nine microbiomes. Each column represents a replicate. I1 and I2 indicate the input compositions for the two independent experimental replicates. In the day 3 and day 10 data sets, the first five columns are from one experimental replicate and the second five are from a second experimental replicate. Blank columns represent replicates that were lost due to contamination. (D) Growth of each of the community members alone (open circles) and in the presence of the community (closed black squares). Each point represents the mean CFU of the taxa, and the error bars represent 1 standard deviation of the mean. Asterisks indicate significant differences between growth alone and growth in the community ($n = 5$, $t$ test, $P < 0.05$).

natural rind cheese and was previously demonstrated to inhibit cheese rind bacterial growth (22). Across isolates of all three taxa, both the 6% NaCl and +*Penicillium* treatments caused a general decrease in total growth across all nine microbiomes, with +*Penicillium* causing stronger growth inhibition (Fig. 4A). Rind microbiomes had variable responses to the two perturbations. The *Penicillium* perturbation caused the most significant shifts in community composition, with six of the nine communities showing significant changes in community composition (Fig. 4B and C). In some communities (C2 and C3), the presence of *Penicillium* caused a major increase in *Brachybacterium* relative abundance. In others (C1, C8, and C9), the presence of *Penicillium* caused an increase in the relative abundance of *Staphylococcus*. The 6% salt treatment caused fewer shifts in community composition, with only two communities (C5 and C6) responding to the higher-salt environment. In both cases, *Brevibacterium* increased in relative abundance.

**Strain-level diversity in synthetic cheese rind communities drives divergence in pigment and aroma production.** Our experiments described above demonstrated that strain-level diversity of cheese rind taxa drove divergence in community composition across the nine microbiomes. Does this divergence lead to the production of cheeses with different properties that could be perceived by consumers? Differences in community composition may not necessarily translate into differences in functional outputs. Many studies of the microbiome have suggested that communities with different compositions may have similar functions due to functional redundancy across community members (48–50). While our comparative genomic analysis described above suggested potential functional differences across the cheese communities, many of the core community functions were conserved, and variation in accessory genes may have little impact on community functions. To determine whether divergence in composition of the synthetic cheese rind communities also translated into differences in functional outputs, we measured two important traits of cheese rind microbiomes: rind color and volatile organic compound (VOC) production.

Cheese rind bacteria define how the cheese appears to customers through the production of cellular pigments such as carotenoids or the secretion of pigmented extracellular metabolites into the curd (27, 51–54). The three bacteria in our model community produced distinct pigments (Fig. 1B), and shifts in their relative abundance were found to be able to translate into changes in rind color. Intraspecific differences in pigment production was also found to drive differences in rind community pigmentation. Using a colorimeter, we measured the rind color after 10 days. The communities showed significantly different color development results (ANOVA $F_{9,39} = 524.9$, $P < 0.0001$), with C3, C4, C6, C7, and C9 having significantly greater a* values (where a* represents one of two chromatic coordinates) than the control, indicating more red pigmentation (Fig. 5A). All communities had significantly greater b* values than the control (ANOVA $F_{9,39} = 139.6$, $P < 0.0001$), with C3 and C4 having the highest values and appearing the most orange (Fig. 5A).

As the rind biofilm decomposes fats, proteins, and other components of the cheese substrate, a diversity of VOCs are produced that are aromatic (55–57). Using headspace sorptive extraction (HSSE) followed by gas chromatography-mass spectrometry (GC-MS) analysis (58, 59), we quantified VOCs produced by each community after 10 days of synthetic community development. Across all nine communities, 248 unique VOCs were detected with significant differences in the mean VOCs per community (Fig. 5B) (ANOVA $F_{8,35} = 28.9$, $P < 0.0001$). The compositions of VOCs across the nine cheese communities were significantly different (Fig. 5C) (PERMANOVA $F = 62.38$, $P < 0.001$) (Table S4). As determined using a SIMPER analysis, the following nine compounds contributed more than 1% to the average overall Bray-Curtis dissimilarity: benzyl methyl ketone (27% contribution; odor = floral/fruity), tetramethylpyrazine (19%; odor = nutty/musty/chocolate/coffee), 2,5-dimethylpyrazine (13%; odor = nutty/musty/chocolate/coffee), trimethylpyrazine (12%; odor = nutty/musty/chocolate/coffee), dimethyl disulfide (9%; odor = sulfurous/cabbage/onion), dimethyl trisulfide (2%; odor = sulfurous/cabbage/onion), 2,6-diethylpyrazine (2%; odor = nutty/musty/choco-

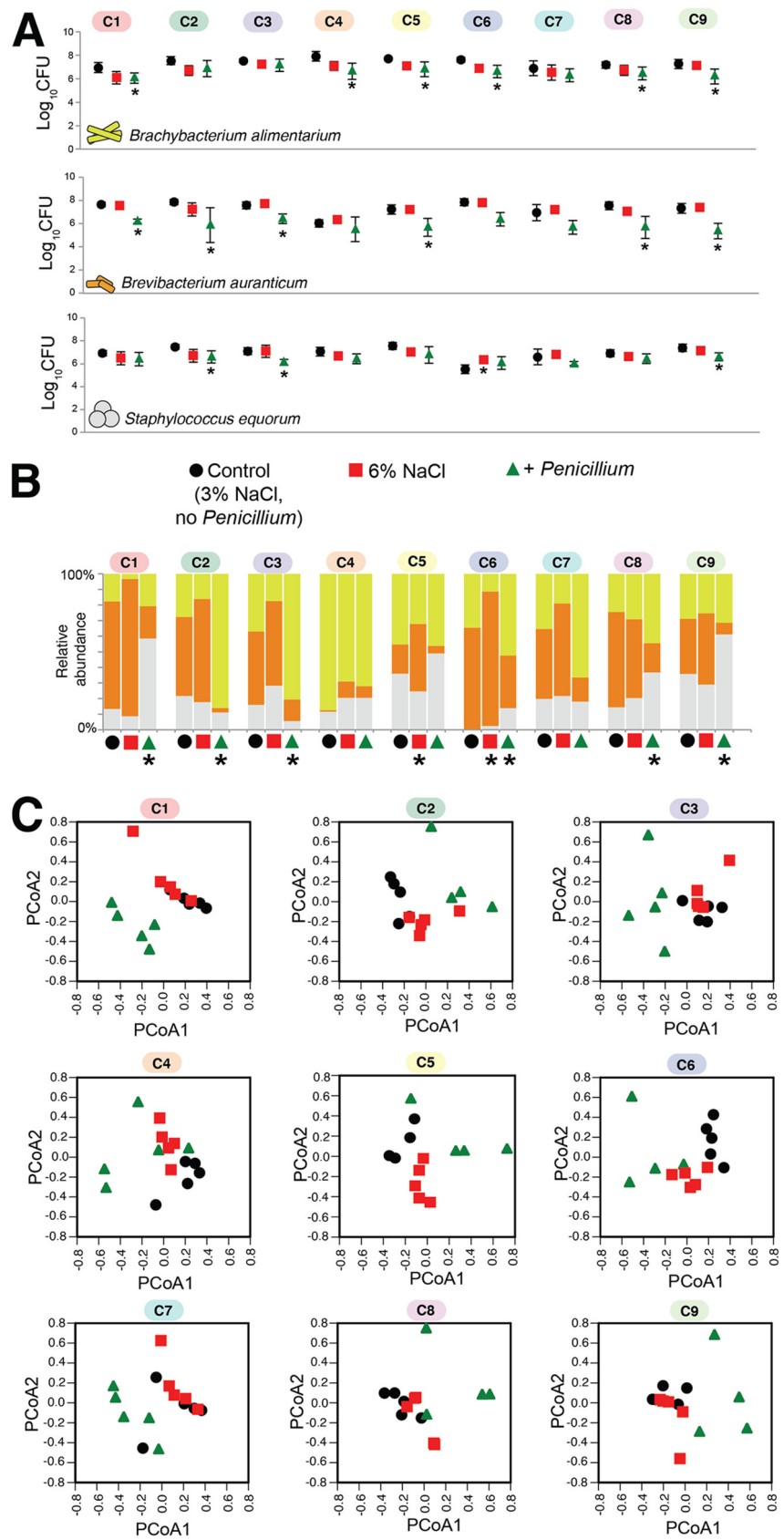

**FIG 4** Response of the nine synthetic cheese rind communities to abiotic and biotic perturbations. (A) Responses of each taxon to abiotic (6% salt) and biotic (*Penicillium*) disturbance. Each point represents

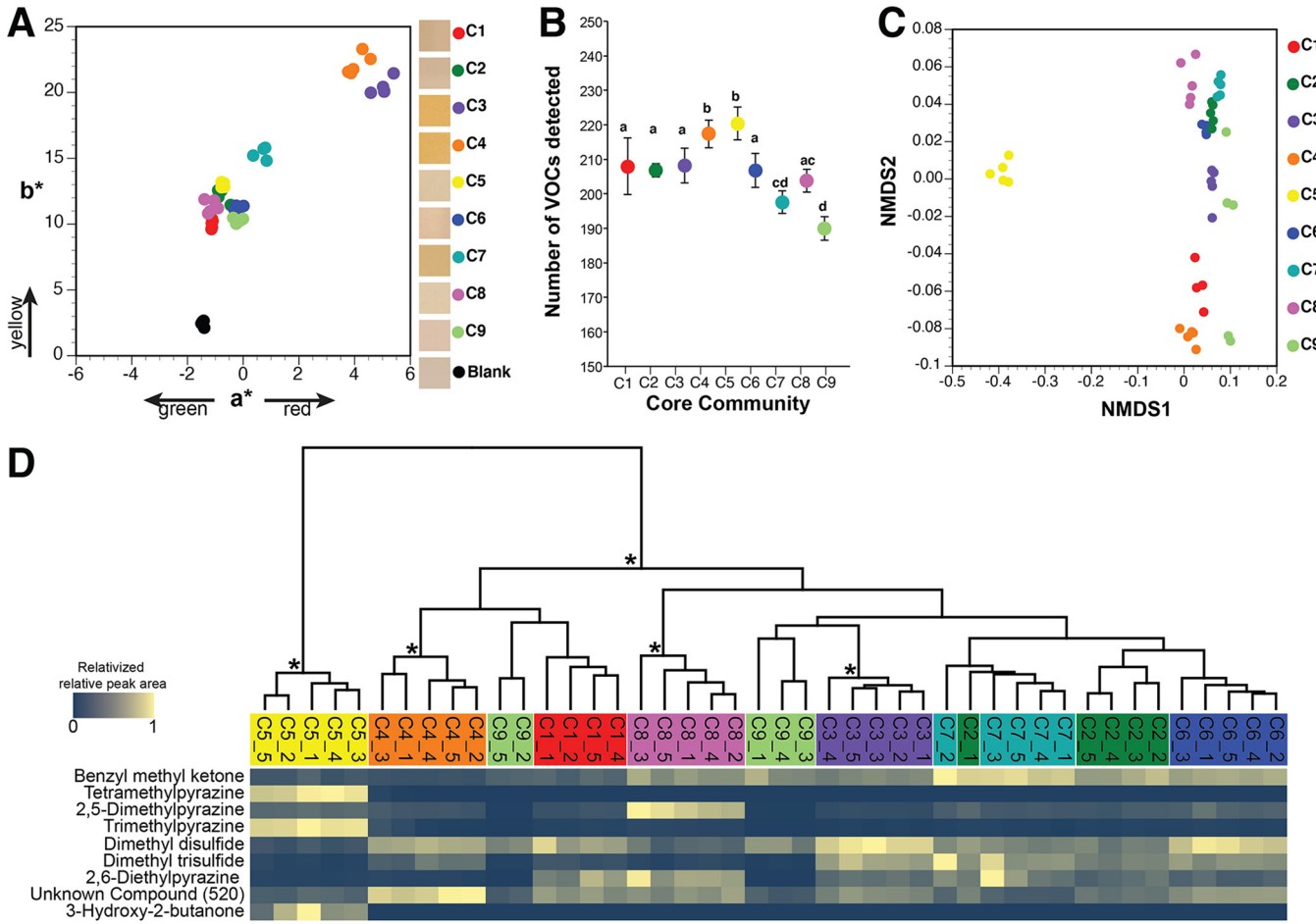

**FIG 5** Functional diversity across nine synthetic cheese rind communities. (A) Color profiles of experimental rind communities after 10 days of rind development. Each dot represents a replicate cheese rind community (*n* = 5). Boxes in the legend show representative photos of the synthetic rind community surface from each community. (B) Total volatile organic compound (VOC) diversity across the nine cheese communities. Each point represents the mean number of VOCs detected in each community, and the error bars represent 1 standard deviation of the mean (*n* = 5). Communities that share the same letter are not significantly different from one another based on the Kruskal-Wallis test (*P* < 0.05). (C) Nonmetric multidimensional scaling of total VOC profiles. Each dot represents a replicate cheese rind community (*n* = 5). (D) Relative abundances of VOCs that contributed the most to the Bray-Curtis dissimilarity across communities (as determined by SIMPER analysis). Because total concentrations of VOCs are highly variable across different compounds, visualization was simplified by relativizing the relative peak area from GC-MS chromatograms within each VOC to the highest concentration detected for that VOC. Data are clustered together by total VOC profiles using a UPGMA tree. Asterisks indicate clusters with >70% bootstrap support. For a full overview of all compounds detected, see Table S4.

late/coffee), unknown compound 520 (1%; odor = unknown), and 3-hydroxy-2-butanone (1%; odor = sweet/buttery/creamy). These VOCs have been detected in other surface-ripened cheeses and result from microbial degradation of fats, proteins, and other components of the cheese substrate (60–65). C5 had the most distinct VOC profile of all communities, with large amounts of tetramethylpyrazine, trimethylpyrazine, and 3-hydroxy-2-butanone and small amounts of the major sulfur compounds, suggesting a nuttier and butterier aroma profile.

The functional differences observed across the nine synthetic cheese rind communities might have resulted from divergent community compositions driven by strain-

**FIG 4** Legend (Continued)
the mean CFU of the taxa in that community at day 10 (*n* = 5), and the error bars represent 1 standard deviation of the mean. Asterisks indicate significant difference in growth compared to control based on the Kruskal-Wallis test (*P* < 0.05). (B) Mean community composition in the three treatments. An asterisk indicates significant difference in community composition compared to control based on PERMANOVA. (C) Principal-coordinate analysis (PCoA) of replicate communities in the three treatments. PCoA data are based on Bray-Curtis dissimilarity of absolute abundances of each community member.

level variations. For example, communities that had final compositions with more *Brevibacterium* would potentially have higher red and yellow pigmentation due to orange carotenoid production by *Brevibacterium*. We do not believe that this explains pigmentation differences across the communities because the two communities with the highest *Brevibacterium* abundances (C1 and C8; Fig. 3C) did not have the highest levels of red or yellow pigmentation (Fig. 5A).

The community-level differences in functions might also have been a result of high intraspecific variation in functional traits (pigment or VOC production) in just one species in the communities, irrespective of the final abundance of that species in the community. For example, when *Brevibacterium* strains differed considerably in their levels of carotenoid production across the nine communities, total community pigment production levels might have varied across the nine communities based solely on trait variation in this one species. Preliminary observations suggest that this may partly explain functional differences of the communities. We noticed that when each of the strains of *Brevibacterium*, *Brachybacterium*, and *Staphylococcus* was grown alone on plate count agar with milk and salt (PCAMS) in the dark (representing the conditions under which the cheese communities were incubated), the results showed considerable variation in pigment production in *Brevibacterium* strains, some pigment variation in *Staphylococcus* strains, and no observable pigment variation in *Brachybacterium* strains (Fig. S3A). Interestingly, the two communities with the highest levels of red and yellow pigmentation (C3 and C4) were also the communities with *Brevibacterium* strains that showed high levels of orange pigmentation (Fig. S3A). Genomes of all the *Brevibacterium* strains harbored predicted genes corresponding to the carotenoid biosynthesis operon previously characterized in *Brevibacterium* species (32, 66, 67), suggesting that regulation of carotenoid genes and not variation in gene content across the strains might explain the variable levels of pigment production. One *Staphylococcus* strain (908_10 from C7) was noticeably yellow in color on CCA (Fig. S3A) and corresponded to one *S. equorum* strain that contained the operon for the carotenoid staphyloxanthin in its genome (Table S4). Further characterization of pigmentation of *Brevibacterium* and *Staphylococcus* on CCA suggests that trait variation of individual species could potentially drive community pigmentation differences. *Brevibacterium* from C3 remained highly pigmented on CCA, while the pigmentation of *Brevibacterium* from C4 observed on PCAMS did not translate to the CCA for unknown reasons (Fig. S3B and C). The *Staphylococcus* strain from C7 remained highly pigmented on CCA relative to the other *Staphylococcus* strains (Fig. S3B and D), which might explain the elevated yellow pigmentation in C7 (Fig. 5A). These preliminary observations suggest that intraspecific variation of individual species across communities could potentially drive differences in community functions.

## DISCUSSION

Using synthetic communities with identical three-member communities isolated from nine distinct cheeses, our work demonstrated the significance of strain-level variation for microbiome assembly and function. Studies of plant and animal communities have demonstrated that intraspecific genetic and phenotypic diversity can impact community assembly and function (68–70). Here, we demonstrate that intraspecific diversity can impact the relative abundance of microbial community members as well as functional outputs of the communities. Many communities did converge on similar compositions despite having substantial variation in accessory gene content. But several communities had substantially different structures and functions even though the initial inocula were identical. Some communities showed relatively even levels of coexistence of the three community members (C2, C3, C5, C7, and C9), while others were dominated by either *Brevibacterium* or *Brachybacterium* (C1, C4, C6, and C8). The divergence was not due to stochastic community assembly across replicates, as we observed highly reproducible community structures across replicate experiments.

The goal of this work was to determine whether strain-level diversity can impact

community dynamics and functions. The limitations imposed by the use of only nine communities made it difficult to pinpoint specific ecological or genetic mechanisms that might underlie the observed differences across the communities. One simple explanation for the dominance of different taxa across the nine rind communities is differences in growth of individual strains. Our experiments comparing growth of individual strains alone versus in the community demonstrated differing growth rates and interactions with the community for each of the three taxa. However, the results do not fully explain the community structures. For example, in C4, where *Brachybacterium* dominated, the levels of growth of a *Brachybacterium* strain alone and in interactions with community members were similar to those seen with other communities where *Brevibacterium* dominated (e.g., C5 and C6). Future work exploring the roles of inhibitory and cooperative interactions will pinpoint specific mechanisms explaining the variable community assembly dynamics of cheese rind microbiomes.

The evolutionary processes that have generated the divergent species and community-level responses of our cheese microbiomes are currently unknown. It is possible that each microbiome has experienced different evolutionary histories in each cheese production environment. As new batches of cheese are introduced to a cave environment, communities may be repeatedly transferred to these new cheeses. This repeated colonization of the cheese substrate could allow rind microbiomes to evolve collectively as a community in the individual production environments (71). Each environment may have unique abiotic selection pressures, including salt concentrations, milk composition, and temperature, that could shape the evolutionary trajectories of these communities. Rind microbiomes could also experience highly divergent biotic environments. For example, these rind communities were isolated from cheeses with variable fungal environments, ranging from yeast to filamentous fungi (22). Future work using experimental evolution to attempt to create divergent communities from an ancestral rind microbiome should help us understand the origins of strain-level diversity in cheese rinds.

Our model communities represent the widespread bacterial taxa found in cheese rinds. We acknowledge that these communities have limitations that might impact translation of our results to other systems. First, our communities harbored only three bacterial species. While some widespread microbiomes have low species diversity (72, 73), many microbiomes have much higher levels of diversity. Would the strain-level diversity of community members have similar impacts in microbiomes with higher taxonomic diversity? With greater potential for higher-order interactions and a higher number of potential functions with increasing species diversity, we predict that increasing diversity might lead to even more highly divergent communities. Our synthetic communities also used a single strain of each species that we isolated from the original nine cheese communities. In constructing our communities, we chose to ignore potential intraspecific variation within each of the nine communities and assumed that the isolated taxa represented the most common genomic type of the species within each community. Metagenomic sequencing studies have identified multiple coexisting strains of the same microbial species (15–18, 20, 34, 74), and these strains might interact with each other and other community members to impact community composition. It would be fascinating to see how inclusion of higher levels of intraspecific diversity within our experimental microbiomes might impact community assembly and function.

The differences that we observed in rind pigmentation and VOC production might have been driven by differences in community composition across the nine communities or by high intraspecific variation of functional traits of individual species or might have been due to unique functions that resulted from interactions between specific strain combinations. Our preliminary observations of high variation in *Brevibacterium* pigmentation across strains suggested that high intraspecific variation of functional traits within one species might be a driver of functional divergence across communities. Variation in levels of carotenoid production across strains of cheese *Brevibacterium* species has been previously reported (27, 32). Additional work needs to be done to understand the potential contributions of strain-specific interspecies interactions in

driving functional differences across these experimental communities. For example, future studies using reciprocal swapping of all strains across all communities might identify interactions between specific strains of each species that might trigger the production of unique pigments or VOCs.

In a previous large amplicon-sequencing study of cheese rind microbiomes (22), we demonstrated that taxonomically identical cheese rind communities could form in very different cheese-making regions. This was surprising given that these cheeses have divergent sensory properties. Many of these differences might be driven by ingredients, length of aging, or other cheese processes. Our current findings suggest that the variability in the qualities of surface-ripened cheeses could also be driven by strain-level differences across the cheese communities. We acknowledge that our lab cheese rinds are not real cheeses and represent only potential patterns of cheese rind community assembly. Our synthetic cheese communities were missing yeasts, filamentous fungi, and other bacteria that are present in cheese rind microbiomes. It is possible that the impacts of strain-level diversity that we observed in our artificial lab communities could translate to larger-scale cheese production. Previous studies of fermented food microbes noted strain-level differences of individual species used in fermented foods (19, 28, 75–78), and other studies demonstrated product-specific and regional diversity in surface-ripened cheeses (79–81). Experimental work performed under more-realistic food production conditions is needed, but our work suggests that this strain-level diversity may be important for understanding and managing the unique identity of cheeses made in specific regions.

More broadly, our work in these synthetic communities may have implications for both the design and management of microbiomes in other systems. First, our work demonstrates that taxonomic profiling of microbiomes may not provide robust predictors of assembly dynamics and functions. Amplicon-based approaches for sequencing microbiomes, such as using 16S rRNA gene sequencing, capture only high-level taxonomic diversity. As we have demonstrated, communities with similar initial species compositions can have very different dynamics. Fortunately, microbiome sequencing studies are moving toward shotgun-metagenomic approaches that could capture the strain-level diversity that we observed across our nine communities (15–18, 20, 34, 74). Our work also suggests that it might be hard to predict microbiome responses to disturbances using taxonomic profiles alone. For example, across individuals that have taxonomically similar microbiomes on their skin, responses to environmental stresses such as the application of antibiotics may depend on the specific strains and genomic content of the communities. Finally, our work suggests that in designing synthetic microbiomes, the choice of the individual "parts" (strains of species) may alter desired outcomes. This may be important for industrial microbial communities where specific strain combinations within communities may be needed to achieve predictable community functions. Strain diversity should also be considered in the design and interpretation of synthetic microbial community experiments used in basic microbiome science. Findings from these systems may not be generalizable if the observed outcomes are unique to the specific strains selected for experiments.

## MATERIALS AND METHODS

**Isolation and maintenance of cheese rind bacterial strains.** Frozen glycerol stocks of communities previously characterized using metagenomic sequencing (22) were plated out on plate count agar with milk and salt (PCAMS) with chloramphenicol (50 mg/liter) to culture bacteria. We specifically selected cheese rind communities that had a high level of abundance of each of the three target genera (*Brachybacterium*, *Brevibacterium*, and *Staphylococcus*) in the metagenomic sequence data from these samples. We initially plated out 50 cheese rinds and isolated putative colonies of *Brachybacterium*, *Brevibacterium*, and *Staphylococcus* from 19 of the samples. We collected isolates from only 19 samples because the remaining samples did not contain colonies with morphologies similar to those of the three target bacteria. A pure culture isolation on a fresh PCAMS plate was made from one colony from each unique morphotype identified within each rind community. *Brachybacterium alimentarium* colonies have medium growth rates, are large and flat, and are yellow-green in color. *Brevibacterium auranticum* colonies are usually slow-growing, medium-sized, and orange. *Staphylococcus equorum* colonies are usually fast-growing, smooth, medium-sized, flat, and either white or light golden in color. Initial

identification of the isolates was done using the 16S rRNA region and primers 27f (82) and 1492r (83) as previously described (54).

**Comparative genomics.** The genome of each bacterial strain was sequenced, assembled, and annotated as we previously described for *Staphylococcus* species (43). Briefly, DNA was extracted using MoBio PowerSoil DNA extraction kits and pure cultures grown for 1 week on PCAMS. Approximately 1 $\mu$g of purified genomic DNA (gDNA) was sheared using a Covaris S220 ultrasonicator to approximately 450-bp lengths and was used as the input for a New England Biolabs NEBNext Ultra DNA library prep kit for Illumina. Libraries were spread across multiple sequencing lanes with other projects and were sequenced using 100-bp, paired-end reads on an Illumina HiSeq 2500 system. Approximately 10 million reads were sequenced for each genome. Failed reads were removed from libraries, and reads were trimmed to remove low-quality bases and were assembled to create draft genomes using the *de novo* assembler in CLC Genomics Workbench 8.0. Assembled genomes were annotated using RAST (84).

Full 16S rRNA gene sequences were extracted from each of the genomes and used to compare 16S gene sequences within each species. These sequences were then aligned using MUSCLE with 16S rRNA gene sequences of closely related type specimens available in NCBI. RAxML 8.2.11 (with GTR GAMMA nucleotide model and 100 bootstrap replicates) was used to create a 16S rRNA gene maximum likelihood phylogeny. To identify phylogenomic relationships within each of the three species as well as between each of the nine communities, we used PanSeq (40) to identify SNPs across the core genome for each of the three species. We first made a maximum likelihood phylogeny for each species using the SNP file for each individual species. We then concatenated the three species SNP files to create a community SNP file. For both the species and community phylogenies, RAxML 8.2.11 (with GTR GAMMA nucleotide model and 100 bootstrap replicates) was used.

To compare the presence and absence of genes across strains and species, core and accessory genes were identified by assigning protein-coding sequences to functionally orthologous groups using the MultiParanoid method of the Pan-Genome Analysis Pipeline (PGAP) (25). Species-to-species orthologs were identified by pairwise strain comparison using BLAST with the following PGAP defaults: a minimum local coverage of 25% of the longer group and a global match of no less than 50% of the longer group, a minimum score value of 50, and a maximum E value of 1e−8. Multistrain orthologs were then found using MultiParanoid (80). Enrichment of SEED subsystem categories in each of the nine communities was determined using Fisher's exact test with false-discovery-rate correction.

**Community assembly assays.** To measure community assembly of the distinct rind communities, approximately 20,000 CFU of each species was inoculated on the surface of 150 $\mu$l of cheese curd agar (3% salt) distributed into replicate wells of a 96-well plate, as previously described (22, 43). Communities were incubated aerobically at 24°C in the dark and were harvested at 3 and 10 days after inoculation, representing early and late community succession, respectively (22). To determine the community composition of individual replicate communities, the community was pestled in 600 $\mu$l of 1$\times$ phosphate-buffered saline, serially diluted, and plated onto PCAMS. PCAMS plates were incubated for a week followed by measuring the abundance of each bacterial species. To measure growth alone, the same density of CFU of each taxon alone was inoculated into wells. Five technical replicates of each community were performed in each of two experimental replicates.

Salt (6%) and fungal (+*Penicillium*) perturbation experiments were conducted using the same community assembly assay but with 6% salt cheese curd agar or with the addition of *Penicillium*. *Penicillium* strain 12, isolated from a natural rind cheese in Vermont, was used in these experiments. We used this strain because it was isolated from a cheese where *Staphylococcus*, *Brachybacterium*, and *Brevibacterium* were also found and because it was used in previous experiments in our lab (43, 47). The exact species identification of this mold is unknown, but it belongs to section *Fasciculata* with other cheese *Penicillium* species. *Penicillium* was inoculated at an initial density of 2,000 CFU. Community composition in these experiments was determined as described above except that cycloheximide was added to the PCAMS plates used for bacterial community isolation to eliminate fungal growth.

**Color and VOC analyses.** To measure rind color and VOC production, we constructed larger versions of each of the nine communities on cheese curd agar poured into petri dishes (60 mm in diameter) to allow a larger sampling area. To construct the rind communities, 600,000 CFU of each species was inoculated across the surface of the cheese curd agar. Synthetic communities were incubated for 10 days in the dark at 24°C followed by color and VOC analyses. For analysis of pigment production by individual strains of each species (see Fig. S3B in the supplemental material), approximately the same density of total CFU used in the three-species communities was spread across the surface of the cheese curd agar.

To measure rind color, we used a CTI spectrocolorimeter (part no. 43237-2). This handheld colorimeter uses the CIELAB color space to quantify both lightness (L*) and two chromatic coordinates (a* and b*). Similar colorimeters have been used to quantify cheese rind color (85). Higher values of a* (a*+) indicate red coloring, while lower values (a*−) indicate green. Higher values of b* (b*+) indicate yellow coloring, while lower values (b*−) indicate blue. Colorimeter readings were taken by placing a 30-mm-diameter petri dish lid upside down on the middle of the surface of the rind and then placing the colorimeter on the petri dish surface. This was done to protect the colorimeter from the sticky rind surface and to avoid cross-contamination across replicates.

Volatiles were collected from synthetic rind communities by headspace sorptive extraction (HSSE) using polydimethylsiloxane (PDMS)-coated magnetic stir-bars. HSSE is an equilibrium-driven enrichment technique in which 10-mm-long-by-0.5-mm-thick stir-bars (Twister; Gerstel) are suspended 1 cm above the sample by placing a magnet on the top side of the collection vessel cover. Five replicates of each culture were sampled for 4 h. After collection, the stir bar was removed and spiked with 10 ppm ethylbenzene-d$_{10}$, an internal standard (obtained from RESTEK). The internal standard was used to

determine the relative concentration of each compound. Organics were introduced into a gas chromatograph/mass spectrometer (GC/MS) by thermal desorption. In addition to the use of the Twister blanks, analysis of the cheese curd agar medium was performed to assess background interferences. Compounds present at equal or higher relative concentrations in the media compared to the samples were removed from the data.

Analyses were performed using an Agilent 7890A/5975C GC/MS equipped with an automated multipurpose sampler (Gerstel). The thermal desorption unit (TDU; Gerstel) provided splitless transfer of the sample from the stir bar into a programmable temperature vaporization inlet (CIS; Gerstel). The TDU was heated from 40°C (0.70 min) to 275°C (3 min) at 600°C/min under 50 ml/min of helium. After 0.1 min, the CIS, operating in solvent vent mode, was heated from −100°C to 275°C (5 min) at 12°C/s. The GC column (HP5-MS; Agilent) (30 m by 250 $\mu$m; pore size, 0.25 $\mu$m) was heated from 40°C (1 min) to 280°C at 5°C/min with 1.2 ml/min of constant helium flow. The MS was scanned from $m/z$ 40 to $m/z$ 350, with the electron ionization (EI) source at 70 eV. A standard mixture of C7 to C30 n-alkanes (Sigma-Aldrich) was used to calculate the retention index (RI) value for each compound in the sample.

Ion Analytics spectral deconvolution software (Gerstel) was used to analyze the GC/MS data (86, 87). A target/nontarget data analysis approach was employed in which previously constructed databases were used to identify target compounds in the sample based on spectral deconvolution of their irons and abundances. Once found, each compound's mass spectrum was subtracted from the peak's total ion current (TIC) signal. Each resulting peak scan was inspected to determine whether the residual ion signals were constant (±20%) or approximated background noise. If constant, the software recorded the retention time, mass spectrum, and 3 to 5 target ions and their relative abundances and entered the data into the database. Finally, sample data were compared to reference compound data in the database, viz., RI and MS (positive identification), or in commercial libraries and literature (tentative identification). Once the data were assigned, the database was annotated to include compound name, CAS number, and RI. If neither positive nor tentative identification was possible (in cases in which the data represented an unknown compound), a numerical identifier was used to identify the compound. The database was annotated to include the GC/MS information described above. In contrast, if peak scans differed (representing unresolved peaks), the software searched for 3 to 5 invariant scans, averaged their spectra, and then subtracted the average spectrum value from the TIC signal. This process was repeated until the residual signal at each scan approximated background noise. If peak signals failed to meet the user-defined criteria described below, no additional information was obtained.

**Statistical analyses.** To determine differences in community composition, permutational multivariate analyses of variance (PERMANOVAs) with Bray-Curtis dissimilarity were used. ANOVA of log-transformed data was used to determine significant differences between total CFU levels across experiments. In cases of unequal variances (represented by growth of individual taxa in perturbations), Kruskal-Wallis tests were used. To determine relationships between relative abundance and growth of individual strains, linear regressions were used. To compare total growth alone to growth in the community, $t$ tests were used. Differences in a* and b* values in the pigmentation assay were determined using ANOVA. To determine differences in VOC composition across the nine communities, PERMANOVAs of Bray-Curtis dissimilarities of relative peak areas were used. A SIMPER analysis of relative peak areas of VOCs was used to identify the contributions of each VOC to Bray-Curtis dissimilarity.

**Data availability.** All genome assemblies have been deposited in NCBI under accession no. GCA_002332225.1, GCA_002332445.1, GCA_002332365.1, GCA_003335435.1, GCA_003335365.1, GCA_002332335.1, GCA_003335395.1, GCA_002332395.1, and GCA_002332325.1 (for *Brevibacterium auranticum*); GCA_003335205.1, GCA_002332305.1, GCA_003335355.1, GCA_003335315.1, GCA_003335415.1, GCA_003335305.1, GCA_002332425.1, GCA_003335265.1, and GCA_003335295.1 (for *Brachybacterium alimentarium*); and GCA_001747895.1, GCA_001747965.1, GCA_001747785.1, GCA_001747865.1, GCA_004143695.1, GCA_001747945.1, GCA_001747805.1, GCA_001747815.1, and GCA_001747825.1 (for *Staphylococcus equorum*) (the accession numbers are also listed in Table S1 in the supplemental material).

## SUPPLEMENTAL MATERIAL

Supplemental material is available online only.

**FIG S1**, TIF file, 2.1 MB.

**FIG S2**, TIF file, 1.6 MB.

**FIG S3**, TIF file, 2 MB.

**TABLE S1**, XLSX file, 0.01 MB.

**TABLE S2**, XLSX file, 2 MB.

**TABLE S3**, XLSX file, 0.1 MB.

**TABLE S4**, XLSX file, 0.2 MB.

## ACKNOWLEDGMENTS

Esther Miller, Grace Cox, Elizabeth Landis, Freddy Lee, and Megan Biango-Daniels provided very helpful feedback on earlier drafts of the manuscript.

This work was supported by a grant from the National Science Foundation (MCB 1715553) to B.E.W. We declare no competing interests with one exception; A.R. developed the Ion Analytics software that is sold by Gerstel.

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
