## [Reviewer comments · mSystems]

Strain-level diversity impacts cheese rind microbiome assembly and function

Brittany Niccum, Erik Kastman, Nicole Kfoury, Albert Robbat, and Benjamin Wolfe

Corresponding Author(s): Benjamin Wolfe, Tufts University

Review Timeline:

Submission Date:	February 19, 2020
Editorial Decision:	March 16, 2020
Revision Received:	May 6, 2020
Editorial Decision:	May 27, 2020
Revision Received:	May 27, 2020
Accepted:	May 28, 2020

Editor: Paul D. Cotter

Reviewer(s): The reviewers have opted to remain anonymous.

Transaction Report:

DOI: <https://doi.org/10.1128/mSystems.00149-20>

Response to Reviews for Previous Submission mSystems00378-19

We appreciate the previous review of our manuscript. Below we respond to each reviewer's comments. The original reviewer comments are in bold. Our responses are in normal text below each comment.

Please note that there is a discrepancy in line numbers between our "Marked-Up Manuscript" file and the Manuscript File without track changes ("Unmarked Manuscript"). This is due to a change in lines that occurs when edits were accepted to make the "Unmarked Manuscript." We refer to the lines numbers in both versions in the responses below.

Reviewer #1 (Comments for the Author):

The authors demonstrate that intraspecific differences among common cheese rind strains lead to differences in microbial community assembly and function (color and volatile organic compounds). I enjoyed reading this work and this intriguing approach to examining intraspecific differences in mixed communities. On the one hand, it is obvious that intraspecific differences will exist and lead to different behaviors in mixed cultures, so the authors' findings are not surprising; on the other hand, this manuscript helps dispel the over-reliance of some investigators on the "core microbiome" concept. I have a few comments and questions about this work:

1. My main concern is that the authors set up a straw man in attacking the "core microbiome" concept. They isolate 3 species that are shared across many different cheeses; however, the abundances of these organisms on these cheeses are vastly different (as are the qualities of the cheeses themselves). So phenotypic differences (e.g., in growth rate, competitiveness) may already be apparent. Differences in abundance of each species (or, better yet, 16S subtype differences within these communities) could reasonably correlate to differences in cheese properties, but the authors seem to imply that the 9 communities are "created equal" simply because these strains were inoculated at even levels. This is a fallacy with "core microbiomes" overall: that they rely on presence-absence information, which can have a tenuous link to functional properties in any environment. I wish the authors would better lay out this problem in the introduction and discussion, and do a better job of showing how these intraspecific differences relate to community differences in the original products.

Based on other comments from this and other reviewers, we have substantially rewritten the introduction. Our rewrite reframes the manuscript with a focus on community-level consequences of strain diversity and moves away from the fallacy of core microbiomes. We believe that this rewriting will help address this reviewer's comment (and comments from other reviewers below)..

We agree that the original products where these strains were isolated have different compositions of the three taxa. Our goal here is not to explain how strain diversity explains the original composition of the cheese rind communities where these strains were isolated from. Instead, we use the intraspecific genomic/phenotypic diversity in these nine cheese communities to test a simple question: does strain-diversity impact community structure and function in microbiomes? The rewriting that we have done should help make this simple question more clear and clarify our overall approach. Our rewriting also more clearly highlights the importance of our work.

2. Do the 3 species isolated here have identical 16S sequences across samples? If not, do the 16S subtypes (e.g., ASVs) relate to the genomic/functional differences that you observe?

We agree that providing information on the 16S sequences of these isolates would help clarify their relationship to described species and illustrate the high similarity of 16S sequences of the strains. We have provided phylogenetic trees of full 16S sequences of each of the strains used in the experiments as well as closely related references sequences from NCBI in the supplemental information (Supplemental Figure S1).

We now note in the Results section (lines 133-135 in the Unmarked Manuscript or lines 222-224 in the Marked-Up Manuscript) that “the 16S rRNA gene region had a 100% pairwise identity within all *Brachy bacterium* and *Staphylococcus* strains and a 99.9% pairwise identity across the *Brevibacterium* strains.”

3. Fig1A demonstrates that there is substantial variation in the relative abundance of these organisms "in the wild". How do the intraspecific differences in community structure relate to the relative abundances of these organisms on the original cheeses of isolation? I.e., do the experimental results reflect the natural composition? Intraspecific variation may control the different abundances of these species/strains in the cheeses themselves, so these phenotypic differences (different rates of growth, color) could be inherent in the products themselves.

This is an excellent question. Unfortunately, we can't make a fair comparison between the original products and the experimental communities we made in the lab because the way composition was measured in both systems was very different. In the original survey of these cheeses, we used 16S amplicon sequencing where the number of amplicon reads was used to estimate species abundances. In our lab experiments, we used culture-based count data of CFUs from communities that were harvested from experimental cheeses. Amplicon count data and CFU data cannot be accurately compared. Moreover (as noted above), the goal of this work was not to explain community structure in the original cheeses.

4. How do differences in rind color/VOC on cheeses relate to the behavior of the individual strains in pure culture? Are the mixed-community differences actually due to different behavior in mixture, or due to purely strain-level differences?

This is an interesting question and we thank the reviewer for bringing it up. Because of the multiple levels of data and experiments in the current manuscript, we did not attempt to tease apart the functions of strains individually versus in the community context. The functional data that we provided (both pigment production and VOCs) were included to demonstrate that the differences in community composition that were observed can translate into differences in functions. We may explore this important question in future studies, but we do not believe that it would improve the rigor, clarity, or scope of the current paper.

5. Have the authors performed any cross-over experiments? E.g., seeing how strains from one community effect the growth of strains in another community. Do we observe different levels of cross-feeding/inhibition?

The reciprocal swapping experiments that the reviewer proposes are an exciting next research direction for this work. We have a series of experiments with our cheese, sourdough, and kombucha systems to address this exact question. The research focus and goals of that work are distinct from the current manuscript and we therefore believe they are beyond the scope of the current study. We look forward to presenting these data in a future manuscript.

minor comments:

line 62 - use "16S rRNA gene" throughout, instead of "16S sequences"

This original reference to 16S sequences was removed with the rewriting of the introduction. We have made sure that all other instances of 16S are "16S rRNA gene" as indicated.

Fig1C does not convey any information, I recommend removing that panel

Thank you for this comment. We have removed this.

Table S1 lists "Brevibacterium linens" but the main text lists B. auranticum

We apologize for this mistake and have corrected this.

Reviewer #2 (Comments for the Author):

The manuscript describes isolates-based simplified communities based on 3 species taken from cheese rinds, reporting the growth behaviour of single members of the community and the response to biotic and abiotic stress along with differences in volatile molecules release in cheese model systems.

As we note below, we are concerned that the author did not understand the experimental design and data. We did not just report the growth behavior of single members of the

community. This is just a small section of our manuscript (Figure 3D and Figure 4A). Most of the manuscript focuses on how strain diversity impacts community-level traits and processes including community composition (relative abundances), responses to perturbations, and functional outputs. All of these data are from when the three species

There are some major concerns.

The aim of this study is not really clear and is somehow diluted in multiple items that are not all explored in enough depth.

We apologize that the aim did not come through clearly in our original manuscript. Our rewrite of the introduction should make our aim more clear (see notes above about our rewriting).

We are not sure what this reviewer means by 'not all explored in enough depth.' This was not clearly explained by the reviewer and this ambiguity means we cannot address their concern.

The authors demonstrate that different strains of the same species but from different sources can give different growth and metabolic responses according to community composition. This information is known to microbial ecologists.

We strongly disagree with this statement. We are concerned that the reviewer did not carefully read our manuscript. We are not saying that individual strains of the same species have different growth rates. What we demonstrate is that communities assembled in an identical environment with different strain compositions have divergent species compositions, responses to perturbations, and functional outputs.

We agree that strain-level variation is known for some microbes. We have cited important reviews that note this in the introduction of our paper. But as we clearly acknowledged in our original submission and even more clearly note in our revised manuscript, most studies of strain-level variation in microbes have been conducted on single species. Our work is novel in that we quantify the ecological significance of strain-level diversity in a multispecies community.

If the information that we present in our manuscript is known, the reviewer should have provided explicit references to previously published papers. These were not provided by the reviewer.

The results of the genomic comparison are not exploited enough and there's no attempt in explaining the differences found in volatile compounds profiles of the model cheese systems.

We did not use the genomes to explain differences in volatile compounds for several reasons:

- 1) The goal of this paper was not to link genomic diversity with volatile production and we did not claim to make those connections. The goal of the genomic sequencing was to demonstrate that there are genetically distinct strains of the three species. The goal of the VOC data is to demonstrate that strain-level differences can impact community function (and not just community composition).

As we acknowledge very clearly in both the original submission and this revision (see lines 322-324 in the Unmarked Manuscript or lines 449-451 in Marked-Up Manuscript)

that we only worked with nine experimental communities and that limits our statistical power to link genomes with community composition or function. In our discussion we very clearly stated “The limited number of core communities (nine) makes it difficult to pinpoint specific ecological or genetic mechanisms that may be underlying the observed differences across the core communities.” There are other excellent studies out there that were designed to address this question. Ours is not one of those studies.

- 2) The genetic basis for the production of the VOCs that were major drivers of differences in VOC profiles (see lines 295-301 in the Unmarked Manuscript or lines 422-428 in Marked-Up Manuscript) are not known for the *Brevibacterium*, *Brachybacterium*, or *Staphylococcus* species used in this study. We would therefore not be able to link VOCs with genes because the genes that produce these VOCs are unknown.
- 3) Differences that we observed in the VOC production would likely be explained by transcriptomic differences (differences in gene expression) not the presence or absence of genes or pathways across strains. Variation in genome content might help explain major differences in VOC production in communities with very different taxonomic compositions, but that was not the context for this work.

Overall the discussion is quite speculative, and the conclusions not fully justified. The overall impact of the results of this study in the specific field is therefore limited.

We would turn this comment back to the reviewer and argue that their comment is not fully justified. What conclusions are not fully justified and why? When peer reviewing manuscripts, please provide specific points to back up your critiques. Vague comments are not helpful for the editor trying to make a decision about the manuscript or the authors who are trying to improve their work.

Specific points

Representativeness of the isolates is questionable and currently not justified. Since the authors want their core microbiome to be representative of a specific cheese/technological condition, such lack of representativeness is an issue.

Again, we do not understand what the reviewer is getting at here because their comments are not specific or fully justified. We do not claim that our core microbiomes are representative of a specific cheese or technological condition. We never claim to have recreated an Époisses, Limburger, or Stilton. We also never claim that everything we observe in our model cheese systems happens in real-world cheese rinds (see detailed explanation of limitations starting with line 349 in the Unmarked Manuscript or line 484 in Marked-Up Manuscript).

Our goal was to create synthetic cheese rind communities that contain some of the most abundant bacterial taxa found in naturally forming cheese rinds. This is similar to many fantastic systems and studies using synthetic communities in many other systems including zebrafish, *Arabidopsis*, mice, etc. None of these studies claims that their work with synthetic communities

explains all of the real dynamics in real-world microbiomes. The goal of their work (as is ours) is to identify general principles of microbiome assembly that are broadly relevant to the system.

How distant were the 9 isolates of the same species were? Some tree or measure of such diversity should be reported.

We have provided phylogenomic trees for each of the individual species used in the communities in Figure S2.

56-58. why is skin given as example here.

We are trying to place our cheese rind model system into a broader context within microbiology. To do this, we cite papers in other realms of microbiology to make broader connections. Moreover, skin is very similar in composition to cheese rind microbiomes: it has low diversity and is largely made up of Firmicutes, Actinobacteria, and some fungi.

71-73. Again, why is this example given. The authors are requested to use appropriate literature that fits the specific subject of the paper.

We provided this example because this is one of the best papers in the field of microdiversity and strain-level diversity that makes the point we are trying to make.

83-84. can this be expressed as percent prevalence? Can this choice be statistically justified?

We have added the exact frequencies that these three genera were detected in lines 94-96 in the Unmarked Manuscript (or lines 124-125 in Marked-Up Manuscript).

92-95. were these species also part of the core microbiome of the rinds? As it is reported, the authors selected core genera on the bases of previous studies but they decided arbitrarily the species to be studied here.

In order to test our hypothesis that strain-level diversity within species can impact cheese rind microbiome diversity and function, we had to select one species within each of the three genera to represent the genus. In our original manuscript, we clearly explained our justification for why we selected these species (lines 109-115 in the Unmarked Manuscript or lines 184-188 in the Marked-Up Manuscript): “The three taxa represent the most common species of the three most abundant bacterial genera in cheese rinds. *Staphylococcus*, *Brevibacterium*, and *Brachybacterium* enter the dairy environment from the raw milk used for cheese production and therefore have the potential to co-occur and adapt to abiotic and biotic conditions within local cheese production facilities. Each species has a distinct colony morphology (**Fig. 1B**) making it easy to track composition in experimental communities.”

281-291. Are all these molecules of microbial origin? Can be all/any of them directly linked to the metabolism of one of the 3 selected species? Given like this, these results have low to no impact as they are not explained.

These VOCs have been identified in previous studies of surface-ripened cheeses. Because the cheese medium background was subtracted out of the analyses, all of these compounds can be reasonably assumed to result from microbial activity. We have added several more lines to this

results section to help explain both of these points (see lines 301-303 in the Unmarked Manuscript or lines 428-430 in the Marked-Up Manuscript).

310-313. This is also due to the design of the study. The genome-level analysis of the communities was not exploited sufficiently in order to work out the strain-level differences that may lead to different behaviours.

See response above (page 4) starting with “The goal of this paper was not to link genomic diversity with volatile production and we did not claim to make those connections...”

323-337. This is quite speculative and not based on the results of this study.

In the discussion, we try to set up a framework for understanding what might generate the diversity that we observed in our study. We indeed are speculating here, but it is thoughtful and grounded speculation to help the reader understand the next steps for this work. We believe that this text is especially helpful for readers of this paper that may not be familiar with the production processes associated with cheeses and have left it in the manuscript.

345-346. this is likely, and such consideration from a food microbial ecology point of view should have discouraged the specific choices made in this experimental design.

We are unsure what the reviewer means here because their comment is not clear.

360-362. Speculative. The reader was not provided with such evidence in the results of this study. Are the difference surely driven by microbial metabolites? Any insights from the gene families?

This is another comment from this reviewer that was ambiguous. We are not sure what the reviewer means by “Are the difference surely drive by microbial metabolites? Any insights from the gene families?” If they mean whether we can determine if gene presences in the genomes predicts volatile organic compounds, we discussed this issue above. We also are very careful in how we frame the applications of our findings in lines 376-378 (lines 527-531 in the Marked-Up Manuscript): “Experimental work in more realistic food production conditions is needed, but our work suggests that putting this strain-level diversity in a community context may be important for understanding and managing the unique identity of cheeses made in specific regions.”

Numbers are not reported in the Methods section. The reader is not provided important info on the number of cheese samples taken into account for the isolation of the strains, the number of different isolates collected (and the number collected/cheese/site), identified, mixed and whose genome was actually sequenced etc. I would suggest to provide a figure or a scheme reporting the experimental design.

Our original survey of cheese rind microbiomes (Wolfe et al. 2014 Cell) was designed to comprehensively describe the diversity of cheese rind microbiomes. That was not at all the goal of the current study. We used the previously collected cheese rinds to isolate the three species used in this paper. In lines 403-411 of the Unmarked Manuscript (lines 565-573 of the Marked-Up Manuscript) we have added the relevant methods details requested by this reviewer.

457. please do not use "experimental cheese" those produced are no cheeses. Authors are strongly advised to use cheese plates or cheese model plates or smt more adherent to what was actually performed.

We are confused by this unclear comment. We did use cheese model plates. We do not know what "smt" means.

398. add reference for the method/primers. Was a 16S sequencing done? It is not clear from the description.

Yes - 16S sequencing was done. We are not sure why this is unclear because we said "Initial identification of the isolates was done using the 16S rRNA region using primers 27f and 1492r." We have added the references for the primers as requested (line 416 of the Unmarked Manuscript or line 578 of the Marked-Up Manuscript) and have cited a previous paper of ours that described the methods for the 16S sequencing (reference 54). The 16S sequencing was simply used for an initial confirmation of ID (that the isolated bacterium was the species we were looking for).

Reviewer #3 (Comments for the Author):

In this manuscript, the investigators identified and isolated core members of cheese rind ecosystems from disparate locations and then re-assembled these microbiomes in vitro, assessing their functional properties over time. They show, perhaps not surprisingly, that strain diversity exists and is an important driver of functional outcomes (in this case, cheese color and aroma). Comments are as follows:

We agree that our results are perhaps not that surprising, but that might be true of so many things in microbiology (or science in general) until they are thoroughly tested. Our work is the first we are aware of to test the significance of strain-level diversity at a community level in microbiomes.

1. The premise is interesting and rather ambitious, and perhaps the only way to address the questions they raise is to simplify the experimental design. This is also a limitation. Specifically, it is not clear how representative these core community isolates are for all of the different cheese samples. For example, the authors state (lines 351-352) the potential coexistence of multiple strains of a species in a community, so how do we know that the C1 Staphylococcus is actually the dominant Staphylococcus of that cheese? Or is that the one strain (among several) that was picked by the luck of the draw?

We have rewritten the manuscript to focus more on the importance of strain-level diversity and less on taxonomically core microbiomes being functional divergent. With the way that we have refocused the manuscript through rewriting, we do not believe that this limitation takes away

from our results. Our goal was to use cheese rinds from disparate locations as sources of unique strains of the same species. We agree and fully acknowledged in the original manuscript that we may not have captured the full genomic and functional diversity within each species from each community sampled. But our sampling of unique strains still provides the important strain-level diversity that we needed to test our hypothesis that divergent strains of the same species can impact community assembly and function.

2. It is not clear from which cheeses the 9 communities were isolated or at least how different they are (nine Muensters from around the world? A variety of hard, soft, aged, young, etc. cheeses?)

We cannot reveal the exact names of each cheese because we promised producers that we would keep the cheese names anonymous. But we agree that more information about the cheese would provide useful context. We have added a column to Table S1 (“Cheese type”) that provides some of the information requested (type of animal where milk was from, raw vs. pasteurized, etc.).

3. The authors could have provided more discussion on microbial cheese ecology relative to the data, including data supportive of their findings (e.g., Schornsteiner et al., 2014, Bokulich and Mills, 2013, Dugat-Bony et al., 2016). They do briefly mention strain differences in fermented foods, in general, but there is vast literature on this and in the strain catalogs that have been described.

The papers that this reviewer notes are important works in the microbial ecology of cheese. However, these papers do not identify or characterize strain-level diversity in cheese microbes so we are unsure how citing these papers would strengthen our manuscript.

4. Not sure about the data points in Figure 3B; the legend indicates $n = 5$, but there appears to be more or less than 5 for some samples.

For the Day 0 inputs, only two points are shown to represent the inputs for each of the two experimental replicates. We apologize for this lack of clarity in the figure legend. We have clarified this in lines 804-805 of the Unmarked Manuscript (lines 1013-1014 of the Marked-Up Manuscript).

March 16, 2020

Prof. Benjamin E. Wolfe
Tufts University
Biology
200 Boston Ave, Suite 4700
Medford, Massachusetts 02155

Re: mSystems00149-20 (Strain-level diversity impacts cheese rind microbiome assembly and function)

Dear Prof. Benjamin E. Wolfe:

The same three reviewers have examined your revised manuscript and noted significant improvements. However I agree with Reviewer 1's concerns and would like you to revise your MS to better acknowledge these concerns. Thus I am considering this a minor modification.

I would also note these reviewers, all eminent scholars in your field, have spent considerable time and effort to review your manuscript and are due your full respect and consideration.

Below you will find the comments of the reviewers.

To submit your modified manuscript, log onto the eJP submission site at <https://msystems.msubmit.net/cgi-bin/main.plex>. If you cannot remember your password, click the "Can't remember your password?" link and follow the instructions on the screen. Go to Author Tasks and click the appropriate manuscript title to begin the resubmission process. The information that you entered when you first submitted the paper will be displayed. Please update the information as necessary. Provide (1) point-by-point responses to the issues raised by the reviewers as file type "Response to Reviewers," not in your cover letter, and (2) a PDF file that indicates the changes from the original submission (by highlighting or underlining the changes) as file type "Marked Up Manuscript - For Review Only."

Please return the manuscript within 60 days; if you cannot complete the modification within this time period, please contact me. If you do not wish to modify the manuscript and prefer to submit it to another journal, please notify me of your decision immediately so that the manuscript may be formally withdrawn from consideration by mSystems.

To avoid unnecessary delay in publication should your modified manuscript be accepted, it is important that all elements you upload meet the technical requirements for production. I strongly recommend that you check your digital images using the Rapid Inspector tool at <http://rapidinspector.cadmus.com/RapidInspector/zmw/>.

Corresponding authors may join or renew ASM membership to obtain discounts on publication fees.

Need to upgrade your membership level? Please contact Customer Service at Service@asmusa.org.

Sincerely,

David Mills

Editor, mSystems

Journals Department
Reviewer comments:

Reviewer #1 (Comments for the Author):

The authors have done a good job reframing their work and the re-write of the introduction better captures the goals and scope of this work. I am mostly satisfied with the authors' response to my review - I think it is fair that many of my initial comments were not critical for this manuscript to be impactful enough for publication, and I am glad to read that my comments have inspired future work in their group.

The authors addressed most of my initial comments adequately, but I am unsatisfied by the authors' response to my comment #4 ("How do differences in rind color/VOC on cheeses relate to the behavior of the individual strains in pure culture? Are the mixed-community differences actually due to different behavior in mixture, or due to purely strain-level differences?") The authors reply that "We may explore this important question in future studies, but we do not believe that it would improve the rigor, clarity, or scope of the current paper." I disagree, and feel that it would immensely improve the rigor and clarity, if not scope - I assert that this test is (1) an important control to more thoroughly examine which species are contributing to the observed phenotypes and (2) is absolutely necessary for the authors to validate their claims of examining strain diversity in a "community context"; i.e., is the sum really greater than the parts? Furthermore, this would address some of reviewer 2's questions about microbial metabolites and relationship between genotype and phenotype.

Reviewer 3 also made some valid points that the authors should consider more carefully, including literature that is valid to the discussion (even if those papers did not evaluate strain-level differences per se, the concepts of product-specific and regional diversity could be contextualized usefully in discussion with the results of the current work)

minor comments:

line 27 - "taxonomically identical microbiomes" is misleading/confusing. I recommend "artificially assembled microbial communities consisting of distinct strains of the same three bacterial species". Calling these simple microbial mixtures "microbiomes" is incorrect here and below (e.g., in the importance statement, discussion, and methods) because a microbiome is defined as the combined microbial contents (and/or genetic components) of a given natural environment; applying this label to a 3-strain model communities constructed in the lab creates confusion about what that model is and what it represents.

line 102-103 - I think it is worth elaborating on what the authors mean by "ecological context" here, especially in light of the reviewer comments and authors' responses regarding the scope and novelty of this work (i.e., how and why mixed-culture tests are a novel aspect of this work).

Reviewer #2 (Comments for the Author):

The authors have partially addressed the comments from the reviewers.
Some replies to the reviewers are unduly disrespectful. This should never happen.

I suggest, again, to delete the expression "experimental cheeses" throughout the manuscript.

Reviewer #3 (Comments for the Author):

The authors have satisfactorily addressed the concerns raised in the initial review

Response to Reviews for Previous Submission mSystems00149-20

We thank all three reviewers for their comments. Below we respond to each reviewer's comments. The original reviewer comments are in bold. Our responses are in normal text below each comment.

Reviewer #1 (Comments for the Author):

The authors have done a good job reframing their work and the re-write of the introduction better captures the goals and scope of this work. I am mostly satisfied with the authors' response to my review - I think it is fair that many of my initial comments were not critical for this manuscript to be impactful enough for publication, and I am glad to read that my comments have inspired future work in their group.

The authors addressed most of my initial comments adequately, but I am unsatisfied by the authors' response to my comment #4 ("How do differences in rind color/VOC on cheeses relate to the behavior of the individual strains in pure culture? Are the mixed-community differences actually due to different behavior in mixture, or due to purely strain-level differences?") The authors reply that "We may explore this important question in future studies, but we do not believe that it would improve the rigor, clarity, or scope of the current paper." I disagree, and feel that it would immensely improve the rigor and clarity, if not scope - I assert that this test is (1) an important control to more thoroughly examine which species are contributing to the observed phenotypes and (2) is absolutely necessary for the authors to validate their claims of examining strain diversity in a "community context"; i.e., is the sum really greater than the parts? Furthermore, this would address some of reviewer 2's questions about microbial metabolites and relationship between genotype and phenotype.

We apologize for not providing a more thoughtful response to this interesting set of experiments proposed by this reviewer. We absolutely agree that it would be really interesting to understand which species are contributing to the functional differences observed in the communities. By comparing the rind color and VOC production of individual species grown alone and in the three-species community combinations, we would be able to determine how species interactions within communities might drive differences in community function.

Due to the COVID-19 pandemic shutting down our lab research and due to personnel shortages in the lab of our chemistry collaborator, we are not able to provide additional VOC data from individual strains to address this question. We were able to collect some qualitative rind color data that begin to address this reviewer's comment. We observe substantial variation in pigmentation in the *Brevibacterium* strains that may drive differences in the community pigmentation profiles. This supports the idea that strain-level differences may drive variation in community-level functions. We have added these results as Figure S3 and as text in the manuscript (Starting on page 14, line 7 in the Results). We also added a paragraph to the discussion (Starting on page 18, line 13) that notes future work is needed to identify whether individual traits versus interactions between species drive differences in community function across the nine communities.

Reviewer 3 also made some valid points that the authors should consider more carefully, including literature that is valid to the discussion (even if those papers did not evaluate

strain-level differences per se, the concepts of product-specific and regional diversity could be contextualized usefully in discussion with the results of the current work)

We now cite these important papers in the discussion (Page 19, line 17).

minor comments:

line 27 - "taxonomically identical microbiomes" is misleading/confusing. I recommend "artificially assembled microbial communities consisting of distinct strains of the same three bacterial species". Calling these simple microbial mixtures "microbiomes" is incorrect here and below (e.g., in the importance statement, discussion, and methods) because a microbiome is defined as the combined microbial contents (and/or genetic components) of a given natural environment; applying this label to a 3-strain model communities constructed in the lab creates confusion about what that model is and what it represents.

We greatly appreciate the reviewer noting this important nuance in terminology and we agree fully with their point. We have changed most instances of "microbiome" in reference to our 3-species model to "synthetic microbial community or "synthetic cheese rind community." We did this in both the main text as well as in subheadings and figure legends.

line 102-103 - I think it is worth elaborating on what the authors mean by "ecological context" here, especially in light of the reviewer comments and authors' responses regarding the scope and novelty of this work (i.e., how and why mixed-culture tests are a novel aspect of this work).

We have reworded this sentence (Page 5, lines 4-5) to make the ecological context more clear. We also hope that the ecological context/consequences of strain-diversity that we are interested in comes through in the third paragraph of the manuscript (starting on Page 3, line 21).

Reviewer #2 (Comments for the Author):

The authors have partially addressed the comments from the reviewers.

Some replies to the reviewers are unduly disrespectful. This should never happen.

We apologize for the disrespect that was perceived by this reviewer in our response. We greatly appreciate the time and effort of all reviewers and never aim to disrespect them with our responses to their comments.

I suggest, again, to delete the expression "experimental cheeses" throughout the manuscript.

We have removed all instances of experimental cheeses as requested and changed to "synthetic cheese rind communities." We have also added additional clarification in a few other places that the work was done with synthetic cheese rind communities in the lab, not real cheeses (see Page 5, lines 17-19).

Reviewer #3 (Comments for the Author):

The authors have satisfactorily addressed the concerns raised in the initial review

We are glad that our edits have satisfied this reviewer. We appreciate their time reviewing our manuscript.

May 27, 2020

Prof. Benjamin E. Wolfe
Tufts University
Biology
200 Boston Ave, Suite 4700
Medford, Massachusetts 02155

Re: mSystems00149-20R1 (Strain-level diversity impacts cheese rind microbiome assembly and function)

Dear Prof. Wolfe (Ben):

I'm happy for this manuscript to be accepted but have selected the minor modifications option for now as there may be a minor typo in line 336 (did you mean *Brachybacterium* rather than *Brevibacterium*?)

Once this has been addressed (as it is better to do so now rather than at the proof stage), I can formally 'accept'

All the best

Paul

To submit your modified manuscript, log onto the eJP submission site at <https://msystems.msubmit.net/cgi-bin/main.plex>. If you cannot remember your password, click the "Can't remember your password?" link and follow the instructions on the screen. Go to Author Tasks and click the appropriate manuscript title to begin the resubmission process. The information that you entered when you first submitted the paper will be displayed. Please update the information as necessary. Provide (1) point-by-point responses to the issues raised by the reviewers as file type "Response to Reviewers," not in your cover letter, and (2) a PDF file that indicates the changes from the original submission (by highlighting or underlining the changes) as file type "Marked Up Manuscript - For Review Only."

Due to the SARS-CoV-2 pandemic, our typical 60 day deadline for revisions will not be applied. I hope that you will be able to submit a revised manuscript soon, but want to reassure you that the journal will be flexible in terms of timing, particularly if experimental revisions are needed. When you are ready to resubmit, please know that our staff and Editors are working remotely and handling submissions without delay. If you do not wish to modify the manuscript and prefer to submit it to another journal, please notify me of your decision immediately so that the manuscript may be formally withdrawn from consideration by mSystems.

To avoid unnecessary delay in publication should your modified manuscript be accepted, it is important that all elements you upload meet the technical requirements for production. I strongly recommend that you check your digital images using the Rapid Inspector tool at <http://rapidinspector.cadmus.com/RapidInspector/zmw/>.

Sincerely,

Paul

Editor, mSystems

Journals Department
Response to Reviews for Previous Submission mSystems00149-20

We thank all three reviewers for their comments. Below we respond to each reviewer's comments. The original reviewer comments are in bold. Our responses are in normal text below each comment.

Reviewer #1 (Comments for the Author):

The authors have done a good job reframing their work and the re-write of the introduction better captures the goals and scope of this work. I am mostly satisfied with the authors' response to my review - I think it is fair that many of my initial comments were not critical for this manuscript to be impactful enough for publication, and I am glad to read that my comments have inspired future work in their group.

The authors addressed most of my initial comments adequately, but I am unsatisfied by the authors' response to my comment #4 ("How do differences in rind color/VOC on cheeses relate to the behavior of the individual strains in pure culture? Are the mixed-community differences actually due to different behavior in mixture, or due to purely strain-level differences?") The authors reply that "We may explore this important question in future studies, but we do not believe that it would improve the rigor, clarity, or scope of the current paper." I disagree, and feel that it would immensely improve the rigor and clarity, if not scope - I assert that this test is (1) an important control to more thoroughly examine which species are contributing to the observed phenotypes and (2) is absolutely necessary for the authors to validate their claims of examining strain diversity in a "community context"; i.e., is the sum really greater than the parts? Furthermore, this would address some of reviewer 2's questions about microbial metabolites and relationship between genotype and phenotype.

We apologize for not providing a more thoughtful response to this interesting set of experiments proposed by this reviewer. We absolutely agree that it would be really interesting to understand which species are contributing to the functional differences observed in the communities. By comparing the rind color and VOC production of individual species grown alone and in the three-species community combinations, we would be able to determine how species interactions within communities might drive differences in community function.

Due to the COVID-19 pandemic shutting down our lab research and due to personnel shortages in the lab of our chemistry collaborator, we are not able to provide additional VOC data from individual strains to address this question. We were able to collect some qualitative rind color data that begin to address this reviewer's comment. We observe substantial variation in pigmentation in the *Brevibacterium* strains that may drive differences in the community pigmentation profiles. This supports the idea that strain-level differences may drive variation in community-level functions. We have added these results as Figure S3 and as text in the manuscript (Starting on page 14, line 7 in the Results). We also added a paragraph to the discussion (Starting on page 18, line 13) that notes future work is needed to identify whether individual traits versus interactions between species drive differences in community function across the nine communities.

Reviewer 3 also made some valid points that the authors should consider more carefully, including literature that is valid to the discussion (even if those papers did not evaluate

strain-level differences per se, the concepts of product-specific and regional diversity could be contextualized usefully in discussion with the results of the current work)

We now cite these important papers in the discussion (Page 19, line 17).

minor comments:

line 27 - "taxonomically identical microbiomes" is misleading/confusing. I recommend "artificially assembled microbial communities consisting of distinct strains of the same three bacterial species". Calling these simple microbial mixtures "microbiomes" is incorrect here and below (e.g., in the importance statement, discussion, and methods) because a microbiome is defined as the combined microbial contents (and/or genetic components) of a given natural environment; applying this label to a 3-strain model communities constructed in the lab creates confusion about what that model is and what it represents.

We greatly appreciate the reviewer noting this important nuance in terminology and we agree fully with their point. We have changed most instances of "microbiome" in reference to our 3-species model to "synthetic microbial community or "synthetic cheese rind community." We did this in both the main text as well as in subheadings and figure legends.

line 102-103 - I think it is worth elaborating on what the authors mean by "ecological context" here, especially in light of the reviewer comments and authors' responses regarding the scope and novelty of this work (i.e., how and why mixed-culture tests are a novel aspect of this work).

We have reworded this sentence (Page 5, lines 4-5) to make the ecological context more clear. We also hope that the ecological context/consequences of strain-diversity that we are interested in comes through in the third paragraph of the manuscript (starting on Page 3, line 21).

Reviewer #2 (Comments for the Author):

The authors have partially addressed the comments from the reviewers.

Some replies to the reviewers are unduly disrespectful. This should never happen.

We apologize for the disrespect that was perceived by this reviewer in our response. We greatly appreciate the time and effort of all reviewers and never aim to disrespect them with our responses to their comments.

I suggest, again, to delete the expression "experimental cheeses" throughout the manuscript.

We have removed all instances of experimental cheeses as requested and changed to "synthetic cheese rind communities." We have also added additional clarification in a few other places that the work was done with synthetic cheese rind communities in the lab, not real cheeses (see Page 5, lines 17-19).

Reviewer #3 (Comments for the Author):

The authors have satisfactorily addressed the concerns raised in the initial review

We are glad that our edits have satisfied this reviewer. We appreciate their time reviewing our manuscript.

May 28, 2020

Prof. Benjamin E. Wolfe
Tufts University
Biology
200 Boston Ave, Suite 4700
Medford, Massachusetts 02155

Re: mSystems00149-20R2 (Strain-level diversity impacts cheese rind microbiome assembly and function)

Dear Prof. Wolfe (Ben)

Your manuscript has been accepted, and I am forwarding it to the ASM Journals Department for publication. For your reference, ASM Journals' address is given below. Before it can be scheduled for publication, your manuscript will be checked by the mSystems senior production editor, Ellie Ghatineh, to make sure that all elements meet the technical requirements for publication. She will contact you if anything needs to be revised before copyediting and production can begin. Otherwise, you will be notified when your proofs are ready to be viewed.

Sincerely,

Paul Cotter
Editor, mSystems

Journals Department
Table S1: Accept
Table S4: Accept
Fig. S1: Accept
Fig. S3: Accept
Table S3: Accept
Table S2: Accept
Fig. S2: Accept